# A method for remotely measuring physical function in large epidemiologic cohorts: Feasibility and validity of a video-guided sit-to-stand test

**Erika Rees-Punia** **\*, Melissa H. Rittase, Alpa V. Patel**

Dept. of Population Science, American Cancer Society, Kennesaw, GA, United States of America

\* Erika.rees-punia@cancer.org

## Abstract

### Introduction

Traditional measures of muscular strength require in-person visits, making administration in large epidemiologic cohorts difficult. This has left gaps in the literature regarding relationships between strength and long-term health outcomes. The aim of this study was to test the feasibility and validity of a video-led, self-administered 30-second sit-to-stand (STS) test in a sub-cohort of the U.S.-based Cancer Prevention Study-3.

### Methods

A video was created to guide participants through the STS test. Participants submitted self-reported scores (n = 1851), and optional video recordings of tests (n = 134). Two reviewers scored all video tests. Means and standard deviations (SD) were calculated for self-reported and video-observed scores. Mean differences (95% confidence intervals (CI)) and Spearman correlation coefficients between self-reported and observed scores were calculated, stratifying by demographic characteristics.

### Results

Participants who uploaded a video reported 14.1 (SD = 3.5) stands, which was not significantly different from the number of stands achieved by the full cohort (13.9 (SD = 4.2), $P$-difference = 0.39). Self-reported and video-observed scores were highly correlated ($\rho$ = 0.97, mean difference = 0.3, 95% CI = 0.1–0.5). There were no significant differences in correlations by sociodemographic factors (all $P$-differences $\geq$0.42).

### Conclusions

This study suggests that the self-administered, video-guided STS test may be appropriate for participants of varying ages, body sizes, and activity levels, and is feasible for implementation within large, longitudinal studies. This video-guided test would also be useful for remote adaptation of the STS test during the COVID-19 pandemic.

**Data Availability Statement:** Some access restrictions apply to the data underlying the findings. Data are from the Cancer Prevention Study 3 and are available from the American

Cancer Society by following the ACS Data Access Procedures (https://www.cancer.org/content/dam/cancer-org/research/epidemiology/cancer-prevention-study-data-access-policies.pdf) for researchers who meet the criteria for access to confidential data.

**Funding:** The author(s) received no specific funding for this work.

**Competing interests:** The authors have declared that no competing interests exist.

## Introduction

There are strong, inverse associations between physical function, including muscular strength, and morbidity and mortality [1–5]. Muscular strength is generally measured in-person via dynamometry [2], one-repetition maximum lifts [4], or grip strength [1]. The equipment, time, and personnel needed for administering these measures makes their application in large epidemiological studies very difficult. This has left numerous gaps in the literature regarding relationships of muscular strength and mobility with other health outcomes, such as cancer prevention or cancer survival, that require a very large study population followed over long periods of time. Therefore, there is a need for measures of strength, mobility, and physical function that can more easily be administered to large, epidemiologic cohorts.

The chair sit-to-stand (STS) test involves the functional movement of rising from a seated position, a maneuver that it is considered essential for independent living [6]. Several studies have explored determinants of STS performance and found that, although ankle, knee, and hip joint strength play a large role in STS ability, balance and other sensorimotor and proprioceptive factors also play a role [7, 8]. Thus, the STS test is viewed as a test of lower limb muscular strength, physical function, and mobility, with scores associated with several shorter-term health outcomes, such as falls [9] and disability [10], in smaller prospective studies.

There are several ways the STS can be administered, though the most common methods are the 5-chair STS test (time to achieve 5 full stands) and the 30-second STS test (number of stands achieved in 30 seconds). Both methods have been extensively tested in a variety of representative and clinical populations, have acceptable test-retest reliability [11], and have moderate to high correlations with leg press performance [11, 12], knee extension force, and ambulatory performance [13]. Both methods have similar psychometric properties, and scores from the two approaches are highly correlated with each other [8].

The aims of the current study were to implement and test the feasibility and validity of a video-led, self-administered STS test in a sub-cohort of Cancer Prevention Study-3 (CPS-3) participants. The 30-second STS was used in the current study to ease video (i.e., not face-to-face) administration.

## Materials and methods

### Study population

CPS-3 is a prospective study of cancer incidence and mortality initiated by the American Cancer Society and is described in detail elsewhere [14]. In short, over 303,000 participants aged 30 to 65 years with no history of cancer provided informed consent and were enrolled between 2006 and 2013. Participants are sent surveys every three years to update exposure information and are followed for incident cancers through linkage with state cancer registries and for cause-specific mortality through linkage with the National Death Index. The Institutional Review Board of Emory University has approved all aspects of CPS-3.

In 2020, 13,052 CPS-3 participants who completed the most recent (2018) survey were sent an email invitation to join a web-based participant portal and the first 3,000 participants to respond were granted access to register for the portal. The STS test was administered via a pre-recorded video through the participant portal along with a brief questionnaire to all 2,976 active registered participants (S1 Fig). Participants were invited via email to complete the survey and STS test over a period of seven weeks.

## STS test administration

A two-minute-long video was created to guide participants through the STS test. The video started with an exercise professional explaining the test; participants were instructed to start seated in a chair with an approximate seat height of 17 inches (at about knee height when standing; 43 centimeters) and without wheels or armrests, then rise from the chair, keeping their arms crossed at their chest, repeatedly throughout the 30-second test. The video continued with the exercise professional administering the test on two participants simultaneously (each moving at different paces, with one completing 9 stands and the other completing 15 stands), and viewers were asked to take the test along with the video participants while a 30-second timer ran in the corner of the video. Participants were instructed to stand/sit as many times as possible while keeping count of the number of times they came to a full stand. Participants were asked to self-report their scores on their first attempt and, if they were able, to record and upload an optional video of themselves taking the test. An additional feedback survey was sent to all registered portal participants which included questions about their experience with the video-administered STS test. Participants who did not complete the STS test within the seven-week period were asked to complete a brief survey regarding their non-response.

## Sociodemographic characteristics

Sociodemographic and health behavior information, including sex (men, women), age (continuous, and collapsed into categories <50, 50–59, and ≥60 years), race/ethnicity (White, Black, Latinx, and other), and body mass index (BMI, calculated with self-reported height and weight) were assessed on prior CPS-3 questionnaires. Details on recent physical activity behaviors (meeting/not meeting aerobic or strength training guidelines) were collected on the 2018 CPS-3 questionnaire using a validated survey, described elsewhere [15, 16].

## Analysis

Participant demographics were calculated as mean (standard deviation (SD)) for continuous and number (percent) for categorical variables. The Wilcoxon rank sum test (continuous variables) or the Fisher exact test (categorical variables) were used to detect significant differences between participants who completed the STS and those who did not.

Two independent reviewers (ER-P, AVP) scored all participant-uploaded video STS tests. Reviewers recorded the number of complete stands and assessed test feasibility by evaluating compliance with several form requirements, including: 1) appropriate chair seat height (approximately 43cm/17in, or about knee height), 2) arms crossed at the chest and not used to assist in standing, 3) contact with the chair made on 'sit' motion, and 4) came to a full stand. Rater agreement was 89% with all differences within ±1 stand; differences were adjudicated until reviewers reached an agreement.

Means (SD) were calculated for self-reported and video-observed number of stands. To understand video STS test validity, mean differences (95% confidence intervals (CI)) and Spearman correlation coefficients between self-reported and observed stands were also calculated; all reported scores were included in analyses. All analyses were performed using SAS v.9.4 (Cary, North Carolina) and an alpha level of 0.05 was considered significant.

## Results

Among the 2,979 CPS-3 portal participants, 1,851 (62%) self-reported an STS score, of which 134 uploaded an accompanying video. Demographics of the full portal cohort and the STS

**Table 1. Demographic characteristics of CPS-3 portal participants by sit-to-stand test completion.**

| | Total portal cohort | Completed STS test | Did not complete STS test | $P_{diff}$[a] |
|---|---|---|---|---|
| | (n = 2,979) | (n = 1,851) | (n = 1,128) | |
| | | N (%) | | |
| **Sex** | | | | 0.080 |
| Women | 1,930 (64.8) | 1,177 (63.6) | 753 (66.8) | |
| Men | 1,049 (35.2) | 674 (36.4) | 375 (33.2) | |
| **Age group** | | | | **<0.001** |
| <50 years | 747 (25.1) | 387 (20.9) | 360 (31.9) | |
| 50–59 years | 899 (30.2) | 518 (28.0) | 381 (33.8) | |
| ≥60 Years | 1,333 (44.7) | 946 (51.1) | 387 (34.3) | |
| **Race/Ethnicity** | | | | 0.052 |
| White non-Latinx | 2,215 (74.4) | 1,408 (76.1) | 807 (71.5) | |
| Black non-Latinx | 124 (4.2) | 74 (4.0) | 50 (4.4) | |
| Latinx | 382 (12.8) | 221 (11.9) | 161 (14.3) | |
| Other | 258 (8.7) | 148 (8.0) | 110 (9.8) | |
| **BMI category** | | | | **<0.001** |
| Underweight | 32 (1.1) | 23 (1.2) | 9 (0.8) | |
| Normal | 1,082 (36.3) | 742 (40.1) | 340 (30.1) | |
| Overweight | 1,035 (34.7) | 652 (35.2) | 383 (34.0) | |
| Obese | 810 (27.2) | 427 (23.1) | 383 (34.0) | |
| Missing | 20 (0.7) | 7 (0.4) | 13 (1.2) | |
| **Physical activity level** | | | | **<0.001** |
| Do not meet aerobic guidelines | 305 (10.2) | 147 (7.9) | 158 (14.0) | |
| Meet aerobic only guidelines | 952 (32.0) | 557 (30.1) | 395 (35.0) | |
| Meet aerobic and strength guidelines | 1,577 (52.9) | 1,067 (57.6) | 510 (45.2) | |
| Missing or Implausible activity data | 145 (4.9) | 80 (4.3) | 65 (5.8) | |

[a]*P* value for difference between those who did and did not complete the STS survey. Boldface indicates statistical significance (p<0.05). CPS-3 = Cancer Prevention Study-3; STS = sit-to-stand test.

video cohort are shown in Table 1. Participants who completed the STS test were generally older, more likely to meet the aerobic and strength training Physical Activity Guidelines for Americans [17] and less likely to be obese than those who did not complete the test.

## Feasibility of video administration

Overall, participants reported an average of 13.9 (SD = 4.1, S2 Fig) stands. Based on responses to the feedback survey, most participants (98.3%) recorded the number of full stands from their first attempt. Among those who reported scores from later attempts (n = 26, 1.7%), reasons included forgetting the score from the first test (n = 7), having an issue/interruption with their first test (n = 16), or feeling they could perform better than their first test (n = 3).

Some participants (n = 423) reported reasons for an incomplete STS, including: forgot (38%), time constraints (15%), lack of interest (18%), physical inability (4%; broken toe, issue with extreme dizziness, torn meniscus, etc.), or other (20%; 'other' write-in responses included technical issues, lack of an appropriate chair, etc.).

Video-observed errors in STS test form are outlined in Table 2. Overall, participants followed the form requirements described in the video (82% with no observed errors), though there were 10 participants (7%) who failed to come to a complete stand once or twice throughout the test and six with a chair that appeared too tall (4%). As far as errors that would produce

**Table 2. Video-observed sit-to-stand test errors.**

| Error description | N (%) |
|---|---|
| Used hands to assist in standing | 2 (1.5) |
| Failed to come to a full stand (knees/hips bent) | 10 (7.4) |
| Failed to make contact with the chair on sit | 1 (0.7) |
| Chair height appeared too high | 6 (4.4) |
| Performed the test for longer than 30 seconds | 2 (1.4) |
| Two or more errors | 3 (2.2) |

completely invalid scores, two participants (1%) counted the number of stands completed in one full minute rather than 30 seconds.

## Validity of self-reported scores compared to video observation

The 134 participants who uploaded an accompanying video reported 14.1 (SD = 3.5) stands, which was not significantly different from the number of stands reported by the 1,717 participants who did not upload a video (13.9 stands, SD = 4.2; *P* difference = 0.39).

Among participants who uploaded a video, the overall mean difference between the self-reported number of stands and the video-observed number of stands was 0.3 (95% CI = 0.1–0.5, Table 3). Overall, the number of self-reported and video-observed stands were highly

**Table 3. Self-reported and video-observed number of stands: Means, mean differences, and spearman correlation coefficients by demographic and behavioral characteristics.**

| | n | Self-reported mean stands (SD) | Video- observed mean stands (SD) | Mean difference (95% CI) | Spearman correlation | $p_{diff}$ [a] |
|---|---|---|---|---|---|---|
| **All** | 134 | 14.1 (3.5) | 13.9 (3.2) | 0.3 (0.1, 0.5) | 0.97 | |
| **Sex** | | | | | | |
| Women | 87 | 13.8 (3.1) | 13.6 (3.1) | 0.2 (0.1, 0.4) | 0.97 | 0.46 |
| Men | 47 | 14.7 (4.1) | 14.3 (3.4) | 0.4 (-0.1, 0.9) | 0.98 | |
| **Age group** | | | | | | |
| <50 years | 39 | 14.8 (3.9) | 14.4 (3.0) | 0.4 (-0.2, 1.1) | 0.94 | 0.53 |
| 50–59 years | 41 | 14.7 (3.4) | 14.5 (3.6) | 0.2 (-0.1, 0.4) | 0.97 | |
| ≥60 years | 54 | 13.2 (3.0) | 13.0 (3.0) | 0.2 (0.03, 0.4) | 0.98 | |
| **Race/Ethnicity** | | | | | | |
| White non-Latinx | 93 | 14.3 (3.5) | 14.0 (3.2) | 0.3 (0.1, 0.6) | 0.97 | 0.84 |
| Black non-Latinx | 7 | 12.4 (4.2) | 12.4 (4.2) | 0.0 (0.0, 0.0) | >0.99 | |
| Latinx | 27 | 14.4 (3.0) | 14.0 (3.0) | 0.3 (0.1, 0.6) | 0.96 | |
| Other | 7 | 13.1 (3.3) | 13.1 (3.3) | 0.0 (0.0, 0.0) | >0.99 | |
| **BMI category** | | | | | | |
| Normal | 74 | 14.6 (3.7) | 14.3 (3.2) | 0.3 (-0.2, 0.6) | 0.97 | 0.94 |
| Overweight | 33 | 13.8 (3.1) | 13.5 (3.2) | 0.3 (0.1, 0.5) | 0.98 | |
| Obese | 23 | 12.8 (2.7) | 12.6 (2.5) | 0.2 (-0.1, 0.4) | 0.98 | |
| **Physical activity level** | | | | | | |
| Do not meet aerobic guidelines | 9 | 12.3 (3.0) | 11.7 (2.9) | 0.7 (0.0, 1.3) | 0.96 | 0.42 |
| Meet aerobic only guidelines | 30 | 12.9 (2.7) | 12.7 (2.7) | 0.2 (-0.2, 0.4) | 0.96 | |
| Meet aerobic and strength guidelines | 88 | 14.8 (3.7) | 14.5 (3.3) | 0.3 (0.0, 0.6) | 0.98 | |
| Missing or implausible data | 7 | 13.0 (1.5) | 13.3 (1.8) | -0.3 (-0.7, 0.2) | 0.92 | |

[a]P for difference in self-report/video-observed mean differences across groups.

correlated ($\rho = 0.97$). There were no significant differences in correlations between the self-reported and video-observed stands across any assessed strata (all $P$ differences $\geq 0.42$)

## Discussion

Traditional measures of muscular strength and mobility require in-person visits with specialized equipment and trained personnel, making them difficult to implement on a large scale. In the current study, we created and tested a self-administered 30-second STS test video to allow for the collection of these data in studies with many participants across large geographic areas. Overall, participants completed the STS test with few errors and almost all of their reported scores were plausible given the average age of the cohort [18].

The in-person requirement for many physical function tests also makes them difficult to implement during the COVID-19 pandemic. Other options for remote delivery include administration via video conferencing, as described in a recent report by Winters-Stone et al. [19]. In that study, the validity of four physical function tests, including the STS test, was examined by comparing scores from tests delivered via video conferencing against in-person test scores. Validity estimates reported in that study were comparable to those in the current study (Pearson product-moment correlation = 0.81 in Winters-Stone, Spearman $\rho = 0.97$ in the current study). The primary difference between the two methods is the requirement of personnel to administer the test. Thus, the video conferencing method may be more appropriate for smaller intervention studies where contact with study personnel is valuable, and the video-guided, self-administered method may be more appropriate in large epidemiologic studies.

In the current study, we were able to confirm the number of stands and critique physical form for a portion of the participants via video observation. There was a high correlation between the number of self-reported and video-observed stands across all sociodemographic groups, with no significant differences in correlations by group. These results suggest that this method of administration may be appropriate for a range of men and women varying in age, body size, and level of activity. Additionally, very few participants made errors that would invalidate their scores. One of the more commonly observed errors was improper chair seat height, which may play an important role in STS test performance, particularly among older adults. One study of fifty-five older adults compared STS scores using a standard-height chair (43cm) versus five randomly ordered seat heights from 80 to 120% of each participant's lower leg length. The mean STS score for the standard chair height was significantly lower than those using chairs from the 120, 110, and 100% conditions ($p < 0.05$), although no significant differences were observed between the standard and the 80% or 90% conditions [20]. In the current study, only 6 out of 134 participants had a seat that appeared to be too high, although researchers could not take exact measurements. Therefore, it may be worth adding an item to the post-STS survey asking participants to confirm their chair height, so that sensitivity analyses restricting to those with a seat 43 cm high could be performed in future studies.

This study is not without limitations. First, participants who completed the STS test were generally older, more active, and less obese than those who did not complete the test. Additionally, 38% of participants did not complete the STS test within the seven-week period; this response could be boosted by utilizing additional reminders and extending the time the survey is available. Finally, it is possible that participants who knew they would be uploading a video recording of their test may be more likely to accurately report their number of stands; however, the distribution of scores among participants who did not include a video was very similar to the distribution among participants with a video, which suggests that scores may have been similarly reported among the two groups. This study benefitted from a large cohort of both men and women that was diverse in terms of race/ethnicity and age.

The ability to implement a measure of lower limb muscular strength and mobility, such as the STS test, within epidemiologic cohorts would facilitate novel studies on associations of physical function with cause-specific mortality, disease incidence, survival, or other quality of life outcomes. Overall, this study suggests that the self-administered, video-guided tool is feasible for implementation within large, longitudinal studies and provides a score that may be useful for understanding participant muscular strength and mobility. This video-guided method would also be appropriate for remote adaptation of the STS test during the COVID-19 pandemic.

## Supporting information

**S1 Fig. Flow of participants.**
(DOCX)

**S2 Fig. Distribution of self-reported stands for the full STS cohort and among those with an accompanying video.**
(DOCX)

## Acknowledgments

The authors express sincere appreciation to all Cancer Prevention Study-3 participants, and to each member of the study and biospecimen management group.

The American Cancer Society funds the creation, maintenance, and updating of the Cancer Prevention Study-3. The views expressed here are those of the authors and do not necessarily represent the American Cancer Society or the American Cancer Society–Cancer Action Network. The authors assume full responsibility for all analyses and interpretation of results.

## Author Contributions

**Conceptualization:** Erika Rees-Punia.

**Data curation:** Erika Rees-Punia.

**Formal analysis:** Melissa H. Rittase.

**Methodology:** Erika Rees-Punia.

**Supervision:** Alpa V. Patel.

**Writing – original draft:** Erika Rees-Punia.

**Writing – review & editing:** Erika Rees-Punia, Melissa H. Rittase, Alpa V. Patel.

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
