## [Decision Letter · Decision Letter 0]

4 Oct 2021

PONE-D-21-28239A method for remotely measuring physical function in large epidemiologic cohorts: feasibility and validity of a video-guided sit-to-stand testPLOS ONE

Dear Dr. Rees-Punia,

Thank you for submitting your manuscript to PLOS ONE. After careful consideration, we feel that it has merit but does not fully meet PLOS ONE’s publication criteria as it currently stands. Therefore, we invite you to submit a revised version of the manuscript that addresses the points raised during the review process.

We look forward to receiving your revised manuscript.

Kind regards,

Yih-Kuen Jan, PhD

Academic Editor

PLOS ONE

Journal Requirements:

"The authors express sincere appreciation to all Cancer Prevention Study-3 participants, and to each member of the study and biospecimen management group. 

The American Cancer Society funds the creation, maintenance, and updating of the Cancer Prevention Study-3. The views expressed here are those of the authors and do not necessarily represent the American Cancer Society or the American Cancer Society – Cancer Action Network. The authors assume full responsibility for all analyses and interpretation of results"

"The American Cancer Society funds the creation, maintenance, and updating of the Cancer Prevention Study-3."

Reviewers' comments:

Reviewer's Responses to Questions

**Comments to the Author**

1. Is the manuscript technically sound, and do the data support the conclusions?

Reviewer #1: Yes

Reviewer #2: Yes

2. Has the statistical analysis been performed appropriately and rigorously? 

Reviewer #1: Yes

Reviewer #2: Yes

3. Have the authors made all data underlying the findings in their manuscript fully available?

Reviewer #1: Yes

Reviewer #2: Yes

4. Is the manuscript presented in an intelligible fashion and written in standard English?

Reviewer #1: Yes

Reviewer #2: Yes

5. Review Comments to the Author

Reviewer #1: The authors attempted to develop an easy and reliable clinical method for measuring lower extremity physical function in a large-scale cohort. Measuring lower limb strength traditionally requires an in-person visit by the professionals, while the opportunity of applying this method on a large-scale population seems pretty low. The current study introduced a video-guided sit-to-stand (STS) test as the possible solution. The results suggest that the self-reported, video-guided STS test might be feasible for implementing in various populations om a large scale. However, some issues regarding methods need to be clarified.

MAJOR COMMENT

This manuscript is well-written and easy to follow. I was curious why you use “p difference” rather than “p value” in this manuscript because p value might be more common in scientific papers.

Introduction

The whole introduction fully justifies the purpose of this study. In my opinion, it would be better to illustrate more about the benefits of improving the performance on STS test.

Materials and methods

First, I did not understand why this study collect the physical activity data. Is there any purpose? Second, why did this study try to detect the difference between participants who complete the STS and the full cohort? Did this analysis answer any research question of this study? Moreover, this study did not contain the discussion about this result. Third, the authors found that the participants who completed the STS test were more likely to fulfill the physical activity recommendation of clinical guidelines. I consider it might be worth discussing why these people would be more willing to complete the STS test (better compliance?)

MINOR COMMENT

Title

Physical function is too vague and broad. I suggest adding lower limb/extremity to specify which part of physical function.

Materials and methods

Line 109: Did you collect the age as continuous variable and divided them into 3 categories? Or the 3 categories are raw data.

Reference

Please make the formats of all references consistent.

Reviewer #2: The topic “A method for remotely measuring physical function in large epidemiologic cohorts: feasibility and validity of a video-guided sit-to-stand test” is interesting. However , some minor concerns were needed to be addressed.

1. What is the content of the a video-guided sit-to-stand test? How the subjects exam their test for self-reported score?

2. and what’s the standard for making the video and how to make sure the subjects understand what they should do by the video’s instruction?

3. What kind of data can be obtained from the video uploaded by the subjects?

4. What kind of physical function information can be applied for the cancer prevention?

5. Due to sit-to-stand test was limited to complete within 30 seconds, is that will narrow the specific group?

6. PLOS authors have the option to publish the peer review history of their article (what does this mean?). If published, this will include your full peer review and any attached files.

Reviewer #1: No

Reviewer #2: No

---

## [Author Response · Author response to Decision Letter 0]

20 Oct 2021

Reviewer #1: The authors attempted to develop an easy and reliable clinical method for measuring lower extremity physical function in a large-scale cohort. Measuring lower limb strength traditionally requires an in-person visit by the professionals, while the opportunity of applying this method on a large-scale population seems pretty low. The current study introduced a video-guided sit-to-stand (STS) test as the possible solution. The results suggest that the self-reported, video-guided STS test might be feasible for implementing in various populations om a large scale. However, some issues regarding methods need to be clarified.

MAJOR COMMENT

This manuscript is well-written and easy to follow. I was curious why you use “p difference” rather than “p value” in this manuscript because p value might be more common in scientific papers.

Response: The p difference is indeed a p value. It is often referred to as p-diff or pdiff. It is the p value from a test of differences. We have a footnote in the table to clarify: “P value for difference between those who did and did not complete the STS survey. Boldface indicates statistical significance (p<0.05).”

Materials and methods

First, I did not understand why this study collect the physical activity data. Is there any purpose? Second, why did this study try to detect the difference between participants who complete the STS and the full cohort? Did this analysis answer any research question of this study? Moreover, this study did not contain the discussion about this result. Third, the authors found that the participants who completed the STS test were more likely to fulfill the physical activity recommendation of clinical guidelines. I consider it might be worth discussing why these people would be more willing to complete the STS test (better compliance?)

Response: We clarified the collection of the physical activity data in the methods: “Sociodemographic and health behavior information, including sex (men, women), age (continuous, and collapsed into categories <50, 50-59, and ≥60 years), race/ethnicity (White, Black, Latinx, and other), and body mass index (BMI, calculated with self-reported height and weight) were assessed on prior CPS-3 questionnaires. Details on recent physical activity behaviors (meeting/not meeting aerobic or strength training guidelines) were collected on the 2018 CPS-3 questionnaire using a validated survey…”

We tested if there was a difference between participants who did the STS test and those who did not to see if certain groups of people might be more likely to do the test (for example, are more physically active people more likely to do the test than physically inactive people?), described here in the methods: “The Wilcoxon rank sum test (continuous variables) or the Fisher exact test (categorical variables) were used to detect significant differences between participants who completed the STS and those who did not.” We discuss the difference between those who did and did not do the test here: “Participants who completed the STS test were generally older, more likely to meet the aerobic and strength training Physical Activity Guidelines for Americans,(17) and less likely to be obese than those who did not complete the test.” Although we do not speculate why, we do describe this information in our limitations section: “This study is not without limitations. First, participants who completed the STS test were generally older, more active, and less obese than those who did not complete the test.”

MINOR COMMENT

Title

Physical function is too vague and broad. I suggest adding lower limb/extremity to specify which part of physical function.

Response: We describe in lines 50-56 that the STS is a test of various aspects of physical function: “Several studies have explored determinants of STS performance and found that, although ankle, knee, and hip joint strength play a large role in STS ability, balance and other sensorimotor and proprioceptive factors also play a role.(7, 8) Thus, the STS test is viewed as a test of lower limb muscular strength, overall physical function, and mobility…”. We hope that including the name of the test in full (“sit-to-stand test”) in the title makes it very clear what test of physical function will be discussed in the paper. 

Materials and methods

Line 109: Did you collect the age as continuous variable and divided them into 3 categories? Or the 3 categories are raw data.

Response: We collected age continuously. We have clarified this point: “Sociodemographic information, including sex (men, women), age (continuous, and collapsed into categories <50, 50-59, and ≥60 years)…”

Reference

Please make the formats of all references consistent.

Response: We have updated the references to follow the journal formatting. 

Reviewer #2: The topic “A method for remotely measuring physical function in large epidemiologic cohorts: feasibility and validity of a video-guided sit-to-stand test” is interesting. However , some minor concerns were needed to be addressed.

1. What is the content of the a video-guided sit-to-stand test? How the subjects exam their test for self-reported score?

Response: The video-guided test is explained in lines 91-100. The participants were instructed to follow this video while taking the test. Then they self-reported their own scores. 

2. and what’s the standard for making the video and how to make sure the subjects understand what they should do by the video’s instruction?

Response: Apologies if we are misunderstanding the comment around the ‘standard for making the video’. We have clarified that the accompanying video upload was optional: “Participants were asked to self-report their scores on their first attempt and, if they were able, to record and upload an optional video of themselves taking the test.”

We were able to assure that the participants understood the STS video instructions by viewing their recorded test videos. Any errors they made in taking the test are outlined in Table 2. Also, we got additional information from the feedback survey. For example, we know that most participants followed the instruction to report the score from their first attempt: “Based on responses to the feedback survey, most participants (98.3%) correctly recorded the number of full stands from their first attempt.”

3. What kind of data can be obtained from the video uploaded by the subjects?

Response: In lines 121-125, we describe how we used the uploaded videos: “Two independent reviewers (ER-P, AVP) scored all participant-uploaded video STS tests. Reviewers recorded the number of complete stands and assessed test feasibility by evaluating compliance with several form requirements, including: 1) appropriate chair seat height (approximately 43cm/17in, or about knee height), 2) arms crossed at the chest and not used to assist in standing, 3) contact with the chair made on ‘sit’ motion, and 4) came to a full stand.” We added the word ‘uploaded’ in case it was not clear that this information pertained to the uploaded videos. 

In other words, we used the videos to collect data on errors in participants’ physical form while they were performing the test.

4. What kind of physical function information can be applied for the cancer prevention?

Response: We have added an additional statement on the ability to use the data collected from this test in chronic disease epidemiology: “The ability to implement a measure of lower limb muscular strength and mobility, such as the STS test, within epidemiologic cohorts would facilitate novel studies on associations of physical function with cause-specific mortality, disease incidence, survival, or other quality of life outcomes.”

5. Due to sit-to-stand test was limited to complete within 30 seconds, is that will narrow the specific group?

Response: Apologies if we are not understanding this comment, but the 30 second sit-to-stand test is a commonly administered test in clinical settings and has been around for decades (we reference papers on this test from as far back as 1995). We did not decide on this time limit ourselves. Further, this length of time has been found to be sufficient for assessing physical function, so we would not propose changing it. We simply developed a way to administer this test remotely via a recorded video.

---

## [Decision Letter · Decision Letter 1]

9 Nov 2021

A method for remotely measuring physical function in large epidemiologic cohorts: feasibility and validity of a video-guided sit-to-stand test

PONE-D-21-28239R1

Dear Dr. Rees-Punia,

We’re pleased to inform you that your manuscript has been judged scientifically suitable for publication and will be formally accepted for publication once it meets all outstanding technical requirements.

Kind regards,

Yih-Kuen Jan, PhD

Academic Editor

PLOS ONE

Additional Editor Comments (optional):

Reviewers' comments:

Reviewer's Responses to Questions

**Comments to the Author**

1. If the authors have adequately addressed your comments raised in a previous round of review and you feel that this manuscript is now acceptable for publication, you may indicate that here to bypass the “Comments to the Author” section, enter your conflict of interest statement in the “Confidential to Editor” section, and submit your "Accept" recommendation.

Reviewer #1: All comments have been addressed

2. Is the manuscript technically sound, and do the data support the conclusions?

Reviewer #1: Yes

3. Has the statistical analysis been performed appropriately and rigorously? 

Reviewer #1: Yes

4. Have the authors made all data underlying the findings in their manuscript fully available?

Reviewer #1: Yes

5. Is the manuscript presented in an intelligible fashion and written in standard English?

Reviewer #1: Yes

6. Review Comments to the Author

Reviewer #1: (No Response)

7. PLOS authors have the option to publish the peer review history of their article (what does this mean?). If published, this will include your full peer review and any attached files.

Reviewer #1: No

---

## [Editor Report · Acceptance letter]

11 Nov 2021

PONE-D-21-28239R1 

A method for remotely measuring physical function in large epidemiologic cohorts: feasibility and validity of a video-guided sit-to-stand test 

Dear Dr. Rees-Punia:

I'm pleased to inform you that your manuscript has been deemed suitable for publication in PLOS ONE. Congratulations! Your manuscript is now with our production department. 

Kind regards, 

on behalf of

Dr. Yih-Kuen Jan 

Academic Editor

PLOS ONE